# Approximating the Permanent by Sampling from Adaptive Partitions

**Jonathan Kuck[1], Tri Dao[1], Hamid Rezatofighi[1], Ashish Sabharwal[2],** and **Stefano Ermon[1]**
[1]Stanford University    [2]Allen Institute for Artificial Intelligence
{kuck,trid,hamidrt,ermon}@stanford.edu, ashishs@allenai.org

## Abstract

Computing the permanent of a non-negative matrix is a core problem with practical applications ranging from target tracking to statistical thermodynamics. However, this problem is also #P-complete, which leaves little hope for finding an exact solution that can be computed efficiently. While the problem admits a fully polynomial randomized approximation scheme, this method has seen little use because it is both inefficient in practice and difficult to implement. We present ADAPART, a simple and efficient method for drawing exact samples from an unnormalized distribution. Using ADAPART, we show how to construct tight bounds on the permanent which hold with high probability, with guaranteed polynomial runtime for dense matrices. We find that ADAPART can provide empirical speedups exceeding 30x over prior sampling methods on matrices that are challenging for variational based approaches. Finally, in the context of multi-target tracking, exact sampling from the distribution defined by the matrix permanent allows us to use the optimal proposal distribution during particle filtering. Using ADAPART, we show that this leads to improved tracking performance using an order of magnitude fewer samples.

## 1 Introduction

The permanent of a square, non-negative matrix $A$ is a quantity with natural graph theoretic interpretations. If $A$ is interpreted as the adjacency matrix of a directed graph, the permanent corresponds to the sum of weights of its cycle covers. If the graph is bipartite, it corresponds to the sum of weights of its perfect matchings. The permanent has many applications in computer science and beyond. In target tracking applications [47, 37, 38, 40], it is used to calculate the marginal probability of measurements-target associations. In general computer science, it is widely used in graph theory and network science. The permanent also arises in statistical thermodynamics [7].

Unfortunately, computing the permanent of a matrix is believed to be intractable in the worst-case, as the problem has been formally shown to be #P-complete [48]. Surprisingly, a fully polynomial randomized approximation scheme (FPRAS) exists, meaning that it is theoretically possible to accurately approximate the permanent in polynomial time. However, this algorithm is not practical: it is difficult to implement and it scales as $O(n^7 \log^4 n)$. Ignoring coefficients, this is no better than exact calculation until matrices of size 40x40, which takes days to compute on a modern laptop.

The problems of sampling from an unnormalized distribution and calculating the distribution's normalization constant (or partition function) are closely related and interreducible. An efficient solution to one problem leads to an efficient solution to the other [30, 28]. Computing the permanent of a matrix is a special instance of computing the partition function of an unnormalized distribution [51]. In this case the distribution is over $n!$ permutations, the matrix defines a weight for each permutation, and the permanent is the sum of these weights.

## 1.1 Contributions

First, we present ADAPART, a novel method for *drawing exact samples from an unnormalized distribution using any algorithm that upper bounds its partition function*. We use these samples to estimate and bound the partition function with high probability. This is a generalization of prior work [25, 32], which showed that a specific bound on the matrix permanent nests, or satisfies a Matryoshka doll like property where the bound recursively fits within itself, for a fixed partitioning of the state space. Our novelty lies in adaptively choosing a partitioning of the state space, which (a) is suited to the particular distribution under consideration, and (b) allows us to use any upper bound or combination of bounds on the partition function, rather than one that can be proven *a priori* to nest according to a fixed partitioning.

Second, we provide a complete instantiation of ADAPART for sampling permutations with weights defined by a matrix, and correspondingly computing the permanent of that matrix. To this end, we identify and use an upper bound on the permanent with several desirable properties, including being computable in polynomial time and being tighter than the best known bound that provably nests.

Third, we empirically demonstrate that ADAPART is both computationally efficient and practical for approximating the permanent of a variety of matrices, both randomly generated and from real world applications. We find that ADAPART can be over 30x faster compared to prior work on sampling from and approximating the permanent. In the context of multi-target tracking, ADAPART facilitates sampling from the optimal proposal distribution during particle filtering, which improves multi-target tracking performance while reducing the number of samples by an order of magnitude.

## 2 Background

The *permanent* of an $n \times n$ non-negative matrix $A$ is defined as $\mathrm{per}(A) = \sum_{\sigma \in S_n} \prod_{j=1}^{n} A(j, \sigma(j))$, where the sum is over all permutations $\sigma$ of $\{1, 2, \ldots, n\}$ and $S_n$ denotes the corresponding symmetric group. Let us define the weight function, or unnormalized probability, of a permutation as $w(\sigma) = \prod_{j=1}^{n} A(j, \sigma(j))$. The permanent can then be written as $\mathrm{per}(A) = \sum_{\sigma \in S_n} w(\sigma)$, which is the partition function (normalization constant) of $w$, also denoted $Z_w$.

We are interested in sampling from the corresponding probability distribution over permutations $p(\sigma) = \frac{w(\sigma)}{\sum_{\sigma' \in S_n} w(\sigma')}$, or more generally from any unnormalized distribution where the exact partition function is unknown. Instead, we will assume access to a function that *upper bounds* the partition function, for instance an upper bound on the permanent. By verifying (at runtime) that this upper bound satisfies a natural 'nesting' property w.r.t. a partition of the permutations, we will be able to guarantee exact samples from the underlying distribution. Note that verification is critical since the 'nesting' property does not hold for upper bounds in general.

In the next few sections, we will consider the general case of any non-negative weight function $w$ over $N$ states (i.e., $w : \mathcal{S} \to \mathbb{R}_{\geq 0}, |\mathcal{S}| = N$) and its partition function $Z_w$, rather than specifically discussing weighted permutations of a matrix and its permanent. This is to simplify the discussion and present it in a general form. We will return to the specific case of the permanent later on.

### 2.1 Nesting Bounds

Huber [25] and Law [32] have noted that upper bounds on the partition function that 'nest' can be used to draw exact samples from a distribution defined by an arbitrary, non-negative weight function. For their method to work, the upper bound must nest according to some fixed partitioning $\mathcal{T}$ of the weight function's state space, as formalized in Definition 1. In Definition 2, we state the properties that must hold for an upper bound to 'nest' according to the partitioning $\mathcal{T}$.

**Definition 1** (Partition Tree). *Let $\mathcal{S}$ denote a finite state space. A partition tree $\mathcal{T}$ for $\mathcal{S}$ is a tree where each node is associated with a non-empty subset of $\mathcal{S}$ such that:*

1. *The root of $\mathcal{T}$ is associated with $\mathcal{S}$.*
2. *If $\mathcal{S} = \{a\}$, the tree $\{a\}$ formed by a single node is a partition tree for $\mathcal{S}$.*
3. *Let $v_1, \cdots, v_k$ be the children of the root node of $\mathcal{T}$, and $S_1, \cdots, S_k$ be their associated subsets of $\mathcal{S}$. $\mathcal{T}$ is a partition tree if $S_i, S_j$ are pairwise disjoint, $\cup_i S_i = \mathcal{S}$, and for each $\ell$ the subtree rooted at $v_\ell$ is a partition tree for $S_\ell$.*

**Definition 2** (Nesting Bounds). *Let $w : \mathcal{S} \to \mathbb{R}_{\geq 0}$ be a non-negative weight function with partition function $Z_w$. Let $\mathcal{T}$ be a partition tree for $\mathcal{S}$ and let $\mathcal{S}_{\mathcal{T}}$ be the set containing the subsets of $\mathcal{S}$ associated with each node in $\mathcal{T}$. The function $Z_w^{\mathrm{UB}}(S) : \mathcal{S}_{\mathcal{T}} \to \mathbb{R}_{\geq 0}$ is a* nesting upper bound *for $Z_w$ with respect to $\mathcal{T}$ if:*

1. *The bound is tight for all single element sets: $Z_w^{\mathrm{UB}}(\{i\}) = w(i)$ for all $i \in \mathcal{S}$.[1]*
2. *The bound 'nests' at every internal node $v$ in $\mathcal{T}$. Let $S$ be the subset of $\mathcal{S}$ associated with $v$. Let $S_1, \cdots, S_k$ be the subsets associated with the children of $v$ in $\mathcal{T}$. Then the bound 'nests' at $v$ if:*

$$\sum_{\ell=1}^{k} Z_w^{\mathrm{UB}}(S_\ell) \leq Z_w^{\mathrm{UB}}(S). \tag{1}$$

## 2.2 Rejection Sampling with a Fixed Partition

Setting aside the practical difficulty of finding such a bound and partition, suppose we are *given* a fixed partition tree $\mathcal{T}$ and a guarantee that $Z_w^{\mathrm{UB}}$ nests according to $\mathcal{T}$. Under these conditions, Law [32] proposed a rejection sampling method to perfectly sample an element, $i \sim \frac{w(i)}{\sum_{j \in \mathcal{S}} w(j)}$, from the normalized weight function (see Algorithm A.1 in the Appendix). Algorithm A.1 takes the form of a rejection sampler whose proposal distribution matches the true distribution precisely—except for the addition of *slack* elements with joint probability mass equal to $Z_w^{\mathrm{UB}}(\mathcal{S}) - Z_w$. The algorithm recursively samples a partition of the state space until the sampled partition contains a single element or slack is sampled. Samples of slack are rejected and the procedure is repeated until a valid single element is returned.

According to Proposition A.1 (see Appendix), Algorithm A.1 yields exact samples from the desired target distribution. Since it performs rejection sampling using $Z_w^{\mathrm{UB}}(S)$ to construct a proposal, its efficiency depends on how close the proposal distribution is to the target distribution. In our case, this is governed by two factors: (a) the tightness of the (nesting) upper bound, $Z_w^{\mathrm{UB}}(S)$, and (b) the tree $\mathcal{T}$ used to partition the state space (in particular, it is desirable for every node in the tree to have a small number of children).

In what follows, we show how to substantially improve upon Algorithm A.1 by utilizing tighter bounds (even if they don't nest *a priori*) and iteratively checking for the nesting condition at runtime until it holds.

## 3 Adaptive Partitioning

A key limitation of using the approach in Algorithm A.1 is that it is painstaking to prove *a priori* that an upper bound nests for a yet unknown weight function with respect to a complete, fixed partition tree. Indeed, a key contribution of prior work [25, 32] has been to provide a proof that a particular upper bound nests for any weight function $w : \{1, \ldots, N\} \to \mathbb{R}_{\geq 0}$ according to a fixed partition tree whose nodes all have a small number of children.

In contrast, we observe that it is nearly trivial to empirically verify *a posteriori* whether an upper bound respects the nesting property for a particular weight function for a particular partition of a state space; that is, whether the condition in Eq. (1) holds for a *particular* choice of $S, S_1, \cdots, S_k$ and $Z_w^{\mathrm{UB}}$. This corresponds to checking whether the nesting property holds at an individual node of a partition tree. If it doesn't, we can refine the partition and repeat the empirical check. We are guaranteed to succeed if we repeat until the partition contains only single elements, but empirically find that the check succeeds after a single call to *Refine* for the upper bound we use.

The use of this adaptive partitioning strategy provides two notable advantages: (a) it frees us to choose *any* upper bounding method, rather than one that can be proven to nest according to a fixed partition tree; and (b) we can customize—and indeed optimize—our partitioning strategy on a per weight function basis. Together, this leads to significant efficiency gains relative to Algorithm A.1.

**Algorithm 1** ADAPART: Sample from a Weight Function using Adaptive Partitioning

**Inputs:**

1. Non-empty state space, $\mathcal{S}$
2. Unnormalized weight function, $w : \mathcal{S} \to \mathbb{R}_{\geq 0}$
3. Family of upper bounds for $w$, $Z_w^{\mathrm{UB}}(S) : \mathcal{D} \subseteq 2^{\mathcal{S}} \to \mathbb{R}_{\geq 0}$
4. Refinement function, $Refine : \mathcal{P} \to 2^{\mathcal{P}}$, where $\mathcal{P}$ is the set of all partitions of $\mathcal{S}$

**Output:** A sample $i \in \mathcal{S}$ distributed as $i \sim \frac{w(i)}{\sum_{i \in \mathcal{S}} w(i)}$.

**if** $\mathcal{S} = \{a\}$ **then** Return $a$
$P = \{\mathcal{S}\}$ ; $ub \leftarrow Z_w^{\mathrm{UB}}(\mathcal{S})$
**repeat**
    Choose a subset $S \in P$ to refine: $\{\{S_1^i, \cdots, S_{\ell_i}^i\}\}_{i=1}^{K} \leftarrow Refine(S)$
    **for all** $i \in \{1, \cdots, K\}$ **do**
        $ub_i \leftarrow \sum_{j=1}^{\ell_i} Z_w^{\mathrm{UB}}(S_j^i)$
    $j \leftarrow \arg\min_i ub_i$ ; $P \leftarrow (P \setminus \{S\}) \cup \{S_1^j, \cdots, S_{\ell_j}^j\}$ ; $ub \leftarrow ub - Z_w^{\mathrm{UB}}(S) + ub_j$
**until** $ub \leq Z_w^{\mathrm{UB}}(\mathcal{S})$
Sample a subset $S_i \in P$ with prob. $\frac{Z_w^{\mathrm{UB}}(S_i)}{Z_w^{\mathrm{UB}}(\mathcal{S})}$, or sample *slack* with prob. $1 - \frac{ub}{Z_w^{\mathrm{UB}}(\mathcal{S})}$
**if** $S_m \in P$ is sampled **then** Recursively call ADAPART $(S_m, w, Z_w^{\mathrm{UB}}, Refine)$
**else** Reject *slack* and restart with the call to ADAPART $(\mathcal{S}, w, Z_w^{\mathrm{UB}}, Refine)$

Algorithm 1 describes our proposed method, ADAPART. It formalizes the adaptive, iterative partitioning strategy and also specifies how the partition tree can be created on-the-fly during sampling without instantiating unnecessary pieces. In contrast to Algorithm A.1, ADAPART does not take a fixed partition tree $\mathcal{T}$ as input. Further, it operates with *any* (not necessarily nesting) upper bounding method for (subsets of) the state space of interest.

Figure 1 illustrates the difference between our adaptive partitioning strategy and a fixed partitioning strategy. We represent the entire state space as a 2 dimensional square. The left square in Figure 1 illustrates a fixed partition strategy, as used by [32]. Regardless of the specific weight function defined over the square, the square is always partitioned with alternating horizontal and vertical splits. To use this fixed partitioning, an upper bound must be proven to nest for any weight function. In contrast, our adaptive partitioning strategy is illustrated by the right square in Figure 1, where we choose horizontal or vertical splits based on the particular weight function. Note that slack is not shown and that the figure illustrates the complete partition trees.

**Fixed vs. Adaptive Partitioning**

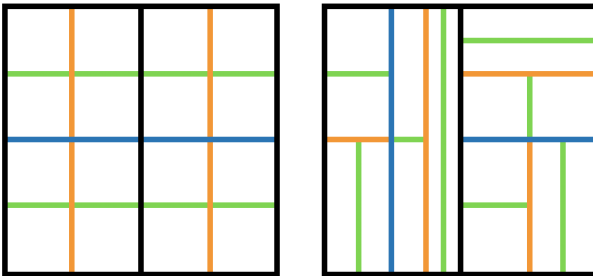

Figure 1: Binary partitioning of a square in the order black, blue, orange, green. Left: each subspace is split in half according to a predefined partitioning strategy, alternating vertical and horizontal splits. Right: each subspace is split in half, but the method of splitting (vertical or horizontal) is chosen adaptively with no predefined order. This figure represents tight upper bounds without slack.

ADAPART uses a function $Refine$, which takes as input a subset $S$ of the state space $\mathcal{S}$, and outputs a collection of $K \geq 1$ different ways of partitioning $S$. We then use a heuristic to decide which one of these $K$ partitions to keep. In Figure 1, $Refine$ takes a rectangle as input and outputs 2 partitionings, the first splitting the rectangle in half horizontally and the second splitting it in half vertically.

ADAPART works as follows. Given a non-negative weight function $w$ for a state space $\mathcal{S}$, we start with the trivial partition $P$ containing only one subset—all of $\mathcal{S}$. We then call $Refine$ on $\mathcal{S}$, which gives $K \geq 1$ possible partitions of $\mathcal{S}$. For each of the $K$ possible partitions, we sum the upper bounds on each subset in the partition, denoting this sum as $ub_i$ for the the $i$-th partition. At this point, we perform a local optimization step and choose the partition $j$ with the tightest (i.e., smallest) upper

bound, $ub_j$. The rest of the $K - 1$ options for partitioning $\mathcal{S}$ are discarded at this point. The partition $P$ is 'refined' by replacing $\mathcal{S}$ with the disjoint subsets forming the $j$-th partition of $\mathcal{S}$.

This process is repeated recursively, by calling *Refine* on another subset $S \in P$, until the sum of upper bounds on all subsets in $P$ is less than the upper bound on $\mathcal{S}$. We now have a valid nesting partition $P$ of $\mathcal{S}$ and can perform rejection sampling. Similar to Algorithm A.1, we draw a random sample from $P \cup \{slack\}$, where each $S_i \in P$ is chosen with probability $\frac{Z_w^{\mathrm{UB}}(S_i)}{Z_w^{\mathrm{UB}}(\mathcal{S})}$, and $slack$ is chosen with the remaining probability. If subset $S_m \in P$ is sampled, we recursively call ADAPART on $S_m$. If $slack$ is selected, we discard the computation and restart the entire process. The process stops when $S_m$ is a singleton set $\{a\}$, in which case $a$ is output as the sample.

ADAPART can be seen as using a greedy approach for optimizing over possible partition trees of $\mathcal{S}$ w.r.t. $Z_w^{\mathrm{UB}}$. At every node, we partition in a way that minimizes the immediate or "local" slack (among the $K$ possible partitioning options). This approach may be sub-optimal due to its greedy nature, but we found it to be efficient and empirically effective. The efficiency of ADAPART can be improved further by tightening upper bounds whenever slack is encountered, resulting in an *adaptive*[2] *rejection sampler* [19] (please refer to Section A.2 in the Appendix for further details).

### 3.1 Estimating the Partition Function

Armed with a method, ADAPART, for drawing exact samples from a distribution defined by a non-negative weight function $w$ whose partition function $Z_w$ is unknown, we now outline a simple method for using these samples to estimate the partition function $Z_w$. The acceptance probability of the rejection sampler embedded in ADAPART can be estimated as

$$\hat{p} = \frac{\text{accepted samples}}{\text{total samples}} \approx p = \frac{Z_w}{Z^{\mathrm{UB}}} \tag{2}$$

which yields $\hat{p} \times Z^{\mathrm{UB}}$ as an unbiased estimator of $Z_w$. The number of accepted samples out of $T$ total samples is distributed as a Binomial random variable with parameter $p = \frac{Z_w}{Z^{\mathrm{UB}}}$. The Clopper–Pearson method [16] gives tight, high probability bounds on the true acceptance probability, which in turn gives us high probability bounds on $Z_w$. Please refer to Section A.3 in the Appendix for the unbiased estimator of $Z_w$ when performing bound tightening as in an adaptive rejection sampler.

## 4 Adaptive Partitioning for the Permanent

In order to use ADAPART for approximating the permanent of a non-negative matrix $A$, we need to specify two pieces: (a) the *Refine* method for partitioning any given subset $S$ of the permutations defined by $A$, and (b) a function that upper bounds the permanent of $A$, as well as any subset of the state space (of permutations) generated by *Refine*.

### 4.1 *Refine* for Permutation Partitioning

We implement the *Refine* method for partitioning an $n \times n$ matrix into a set of $K = n$ different partitions as follows. One partition is created for each column $i \in \{1, \ldots, n\}$. The $i$-th partition of the $n!$ permutations contains $n$ subsets, corresponding to all permutations containing a matrix element, $\sigma^{-1}(i) = j$ for $j \in \{1, \ldots, n\}$. This is inspired by the fixed partition of Law [32, pp. 9-10], modified to choose the column for partitioning adaptively.

### 4.2 Upper Bounding the Permanent

There exists a significant body of work on estimating and bounding the permanent (cf. an overview by Zhang [52]), on characterizing the potential tightness of upper bounds [21, 42], and on improving upper bounds [26, 44, 45, 46]. We use an upper bound from Soules [46], which is computed as follows. Define $\gamma(0) = 0$ and $\gamma(k) = (k!)^{1/k}$ for $k \in \mathbb{Z}_{\geq 1}$. Let $\delta(k) = \gamma(k) - \gamma(k-1)$. Given a matrix $A \in \mathbb{R}^{n \times n}$ with entries $A_{ij}$, sort the entries of each row from largest to smallest to obtain $a_{ij}^*$,

where $a_{i1}^* \geq \cdots \geq a_{in}^*$. This gives the upper bound,

$$\text{per}(A) \leq \prod_{i=1}^{n} \sum_{j=1}^{n} a_{ij}^* \delta(j). \tag{3}$$

If the matrix entries are either 0 or 1, this bound reduces to the Minc-Brègman bound [36, 10]. This upper bound has many desirable properties. It can be efficiently computed in polynomial time, while tighter bounds (also given by [46]) require solving an optimization problem. It is significantly tighter than the one used by Law [32]. This is advantageous because the runtime of ADAPART scales linearly with the bound's tightness (via the acceptance probability of the rejection sampler).

Critically, we empirically find that this bound never requires a second call to *Refine* in the repeat-until loop of ADAPART. That is, in practice we always find at least one column that we can partition on to satisfy the nesting condition. This bounds the number of subsets in a partition to $n$ and avoids a potentially exponential explosion. This is fortuitous, but also interesting, because this bound (unlike the bound used by Law [32]) does not nest according to any predefined partition tree for all matrices.

### 4.3 Dense Matrix Polynomial Runtime Guarantee

The runtime of ADAPART is bounded for dense matrices as stated in Proposition 1. Please refer to Section A.4 in the Appendix for further details.

**Proposition 1.** *The runtime of* ADAPART *is* $O(n^{1.5+.5/(2\gamma-1)})$ *for matrices with $\gamma n$ entries in every row and column that all take the maximum value of entries in the matrix, as shown in Algorithm A.2.*

## 5 Related Work on Approximating the Permanent

The fastest exact methods for calculating the permanent have computational complexity that is exponential in the matrix dimension [41, 6, 4, 20]. This is to be expected, because computing the permanent has been shown to be #P-complete [48]. Work to approximate the permanent has thus followed two parallel tracks, sampling based approaches and variational approaches.

The sampling line of research has achieved complete (theoretical) success. Jerrum et al. [29] proved the existence of a fully polynomial randomized approximation scheme (FPRAS) for approximating the permanent of a general non-negative matrix, which was an outstanding problem at the time [11, 27]. An FPRAS is the best possible solution that can be reasonably hoped for since computing the permanent is #P-complete. Unfortunately, the FPRAS presented by [29] has seen little, if any, practical use. The algorithm is both difficult to implement and slow with polynomial complexity of $O(n^{10})$, although this complexity was improved to $O(n^7 \log^4 n)$ by Bezáková et al. [8].

In the variational line of research, the Bethe approximation of the permanent [24, 49] is guaranteed to be accurate within a factor of $2^{n/2}$ [2]. This approach uses belief propagation to minimize the Bethe free energy as a variational objective. A closely related approximation, using Sinkhorn scaling, is guaranteed to be accurate within a factor of $2^n$ [23]. The difference between these approximations is discussed in Vontobel [49]. The Sinkhorn based approximation has been shown to converge in polynomial time [33], although the authors of [24] could not prove polynomial convergence for the Bethe approximation. Aaronson and Hance [1] build on [22] (a precursor to [23]) to estimate the permanent in polynomial time within additive error that is exponential in the largest singular value of the matrix. While these variational approaches are relatively computationally efficient, their bounds are still exponentially loose.

There is currently a gap between the two lines of research. The sampling line has found a theoretically ideal FPRAS which is unusable in practice. The variational line has developed algorithms which have been shown to be both theoretically and empirically efficient, but whose approximations to the permanent are exponentially loose, with only specific cases where the approximations are good [24, 13, 14]. Huber [25] and Law [32] began a new line of sampling research that aims to bridge this gap. They present a sampling method which is straightforward to implement and has a polynomial runtime guarantee for dense matrices. While there is no runtime guarantee for general matrices, their method is significantly faster than the FRPAS of [29] for dense matrices. In this paper we present a novel sampling algorithm that builds on the work of [25, 32]. We show that ADAPART leads to significant empirical speedups, further closing the gap between the sampling and variational lines of research.

# 6 Experiments

In this section we show the empirical runtime scaling of ADAPART as matrix size increases, test ADAPART on real world matrices, compare ADAPART with the algorithm from Law [32] for sampling from a fixed partition tree, and compare with variational approximations [24, 2, 23]. Please see section A.5 in the Appendix for additional experiments verifying that the permanent empirically falls within our high probability bounds.

## 6.1 Runtime Scaling and Comparison with Variational Approximations

To compare the runtime performance of ADAPART with Law [32] we generated random matrices of varying size. We generated matrices in two ways, by uniformly sampling every element from $[0, 1)$ (referred to as 'uniform' in plots) and by sampling $\lfloor \frac{n}{k} \rfloor$ blocks of size $k$x$k$ and a single $n \bmod k$ block along the diagonal of an $n$x$n$ matrix (with all other elements set to 0, referred to as 'block diagonal' in plots). Runtime scaling is shown in Figure 2. While ADAPART is faster in both cases, we observe the most time reduction for the more challenging, low density block diagonal matrices. For reference a Cython implementation of Ryser's algorithm for exactly computing the permanent in exponential time [41] requires roughly 1.5 seconds for a 20x20 matrix.

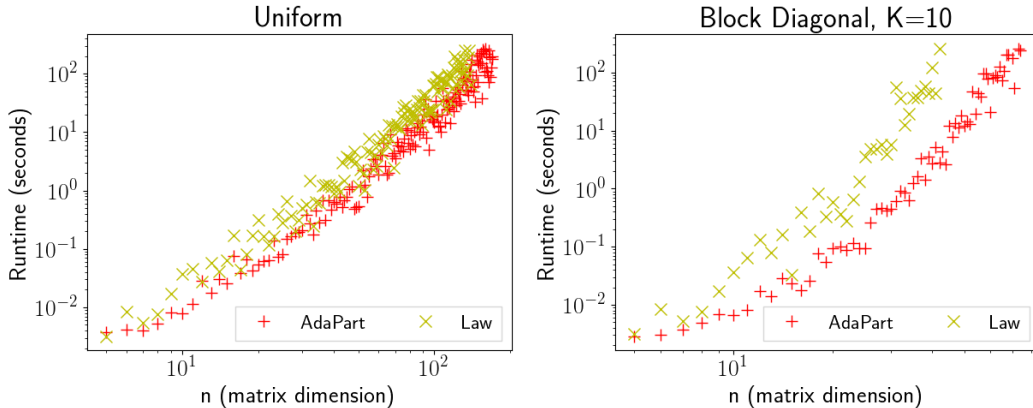

Figure 2: Log-log plot of mean runtime over 5 samples against $n$ (matrices are of size $n \times n$).

To demonstrate that computing the permanent of these matrices is challenging for variational approaches, we plot the bounds obtained from the Bethe and Sinkhorn approximations in Figure 3. Note that the gap between the variational lower and upper bounds is exponential in the matrix dimension $n$. Additionally, the upper bound from Soules [46] (that we use in ADAPART) is frequently closer to the exact permanent than all variational bounds.

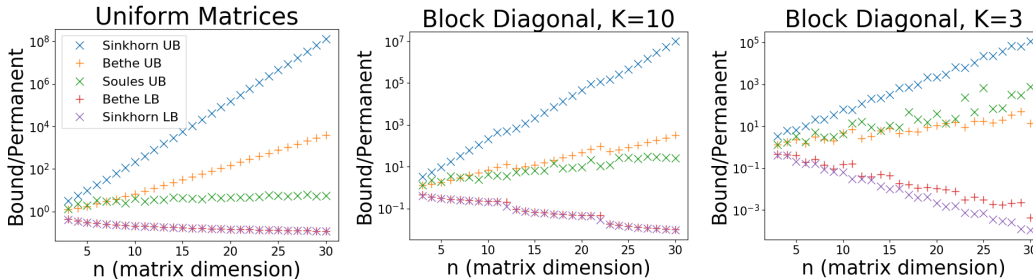

Figure 3: Bounds on the permanent given by the Bethe approximation [24, 2], the Sinkhorn approximation [23], and the upper bound we use from Soules [46].

## 6.2 Matrices from Real-World Networks

In Table 1 we show the performance of our method on real world problem instances. In the context of directed graphs, the permanent represents the sum of weights of *cycle covers* (i.e., a set of disjoint directed cycles that together cover all vertices of the graph) and defines a distribution over cycle covers. Sampling cycle covers is then equivalent to sampling permutations from the distribution defined by the permanent. We sampled 10 cycle covers from distributions arising from graphs[3] in the fields of cheminformatics, DNA electrophoresis, and power networks and report mean runtimes in Table 1. Among the matrices that did not time out, ADAPART can sample cycle covers 12 - 32x faster than the baseline from Law [32]. We used 10 samples from ADAPART to compute bounds on the permanent that are tight within a factor of 5 and hold with probability .95, shown in the ADAPART sub-columns of Table 1 (we show the natural logarithm of all bounds). Note that we would get comparable bounds using the method from [32] as it is also produces exact samples. For comparison we compute variational bounds using the method of [23], shown in the 'Sinkhorn' sub-columns. Each of these bounds was computed in less than .01 seconds, but they are generally orders of magnitude looser than our sampling bounds. Note that our sampling bounds can be tightened arbitrarily by using more samples at the cost of additional (parallel) computation, while the Sinkhorn bounds cannot be tightened. We do not show bounds given by the Bethe approximation because the matlab code from [24] was very slow for matrices of this size and the c++ code does not handle matrices with 0 elements.

| Model Information | | | Sampling Runtime (sec.) | | Lower Bounds | | Upper Bounds | |
|---|---|---|---|---|---|---|---|---|
| Network Name | Nodes | Edges | ADAPART | Law [32] | ADAPART | Sinkhorn | ADAPART | Sinkhorn |
| ENZYMES-g192 | 31 | 132 | **4.2** | 52.9 | **19.3** | 17.0 | **20.8** | 38.5 |
| ENZYMES-g230 | 32 | 136 | **3.3** | 55.5 | **19.8** | 17.2 | **21.3** | 39.4 |
| ENZYMES-g479 | 28 | 98 | **1.8** | 45.1 | **12.3** | 10.9 | **13.8** | 30.3 |
| cage5 | 37 | 196 | **6.1** | TIMEOUT | **-20.2** | -29.2 | **-18.7** | -3.6 |
| bcspwr01 | 39 | 46 | **4.2** | 74.8 | **18.7** | 13.2 | **20.1** | 40.3 |

Table 1: Runtime comparison of our algorithm (ADAPART) with the fixed partitioning algorithm from Law [32] and bound tightness comparison of ADAPART with the Sinkhorn based variational bounds from [23] (logarithm of bounds shown). Best values are in **bold**.

## 6.3 Multi-Target Tracking

The connection between measurement association in multi-target tracking and the matrix permanent arises frequently in tracking literature [47, 37, 38, 40]. It is used to calculate the marginal probability that a measurement was produced by a specific target, summing over all other joint measurement-target associations in the association matrix. We implemented a Rao-Blackwellized particle filter that uses ADAPART to sample from the optimal proposal distribution and compute approximate importance weights (see Section A.6 in the Appendix).

We evaluated the performance of our particle filter using synthetic multi-target tracking data. Independent target motion was simulated for 10 targets with linear Gaussian dynamics. Each target was subjected to a unique spring force. As baselines, we evaluated against a Rao-Blackwellized particle filter using a sequential proposal distribution [43] and against the standard multiple hypothesis tracking framework (MHT) [39, 15, 31]. We ran each method with varying numbers of particles (or tracked hypothesis in the case of MHT) and plot the maximum log-likelihood of measurement associations among sampled particles in Figure 4. The mean squared error over all inferred target locations (for the sampled particle with maximum log-likelihood) is also shown in Figure 4. We see that by sampling from the optimal proposal distribution (blue x's in Figure 4) we can find associations with larger log-likelihood and lower mean squared error than baseline methods while using an order of magnitude fewer samples (or hypotheses in the case of MHT).

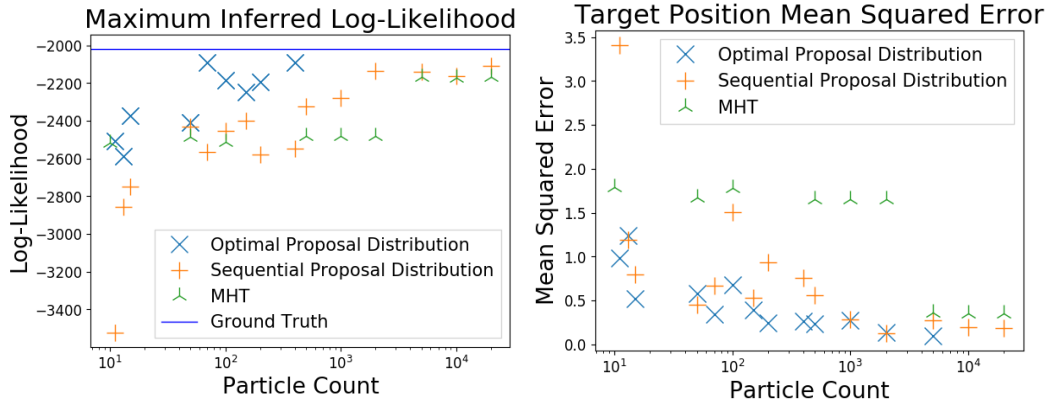

Figure 4: Multi-target tracking performance comparison. Left: maximum log-likelihoods among sampled particles (or top-$k$ hypotheses for the MHT baseline). Right: mean squared error over all time steps and target locations.

# 7 Conclusion and Future Work

Computing the permanent of a matrix is a fundamental problem in computer science. It has many applications, but exact computation is intractable in the worst case. Although a theoretically sound randomized algorithm exists for *approximating* the permanent in polynomial time, it is impractical. We proposed a general approach, ADAPART, for drawing exact samples from an unnormalized distribution. We used ADAPART to construct high probability bounds on the permanent in provably polynomial time for dense matrices. We showed that ADAPART is significantly faster than prior work on both dense and sparse matrices which are challenging for variational approaches. Finally, we applied ADAPART to the multi-target tracking problem and showed that we can improve tracking performance while using an order of magnitude fewer samples.

In future work, ADAPART may be used to estimate general partition functions if a general upper bound [50, 34, 35] is found to nest with few calls to $Refine$. The matrix permanent specific implementation of ADAPART may benefit from tighter upper bounds on the permanent. Particularly, a computationally efficient implementation of the Bethe upper bound [24, 2] would yield improvements on sparse matrices (see Figure 3), which could be useful for multi-target tracking where the association matrix is frequently sparse. The 'sharpened' version of the bound we use (Equation 3), also described in [46], would offer performance improvements if the 'sharpening' optimization problem can be solved efficiently.

**Acknowledgements**

Research supported by NSF (#1651565, #1522054, #1733686), ONR (N00014-19-1-2145), AFOSR (FA9550- 19-1-0024), and FLI.

## Footnotes

[1]This requirement can be relaxed by defining a new upper bounding function that returns $w(i)$ for single element sets and the upper bound which violated this condition for multi-element sets.

[2]The use of 'adaptive' here is to connect this section with the rejection sampling literature, and is unrelated to 'adaptive' partitioning discussed earlier.

[3]Matrices available at http://networkrepository.com.

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
