[Supplementary Material]


# A Appendix

## A.1 Proof of Correctness for Sampling with a Fixed Partition

Algorithm A.1 specifies a method for sampling from a weight function given a fixed partition tree and a bound that provably nests. Its proof of correctness is given in Proposition A.1. Note that a simple property that follows from recursively applying the definition of a nesting bound is that $\sum_{i \in \mathcal{S}} w(i) \leq Z_w^{\text{UB}}(\mathcal{S})$. More generally, given any node $v$ in $\mathcal{T}$ associated with the subset $S_v \subseteq \mathcal{S}$, we have $\sum_{i \in S_v} w(i) \leq Z_w^{\text{UB}}(S_v)$.

**Proposition A.1** (Huber [25], Law [32]). *Algorithm A.1 samples an element $i \in \mathcal{S}$ from the normalized weight function $i \sim \frac{w(i)}{\sum_{j \in \mathcal{S}} w(j)}$.*

*Proof.* The probability of sampling leaf node $v_i$ at depth $d$ in the partition tree, with ancestors $v_{d-1}^a$, $\ldots$, $v_0^a$ (where $v_{d-1}^a$ is the parent node of $v_i$ and $v_0^a$ is the root node) and associated ancestor subsets $S_{d-1}^a, \ldots, S_0^a$ is

$$\frac{1}{p_{accept}} \times \frac{Z_w^{UB}(S_1^a)}{Z_w^{UB}(S_0^a)} \times \frac{Z_w^{UB}(S_2^a)}{Z_w^{UB}(S_1^a)} \times \cdots \times \frac{Z_w^{UB}(S_d^a)}{Z_w^{UB}(S_{d-1}^a)}$$
$$= \frac{1}{p_{accept}} \times \frac{Z_w^{UB}(S_d^a)}{Z_w^{UB}(S_0^a)} = \frac{Z_w^{UB}(\mathcal{S})}{Z_w} \times \frac{w(i)}{Z_w^{UB}(\mathcal{S})} = \frac{w(i)}{\sum_{i \in \mathcal{S}} w(i)}$$

$\square$

---

**Algorithm A.1** Sample from a Normalized Weight Function

**Inputs:**

1. Non-empty state space $\mathcal{S} = \{1, \ldots, N\}$
2. Partition tree $\mathcal{T}$ of $\mathcal{S}$
3. Unnormalized weight function $w : \mathcal{S} \to \mathbb{R}_{\geq 0}$
4. Nesting upper bound $Z_w^{\text{UB}}(S)$ for $w$ with respect to $\mathcal{T}$

**Output:** A sample $i \in \mathcal{S}$ distributed as $i \sim \frac{w(i)}{\sum_{j \in \mathcal{S}} w(j)}$.

**Algorithm:**

1. Set $v$ to the root node of $\mathcal{T}$ and $S = \mathcal{S}$.
2. Sample a child of $v$ (denoted $v_1, \ldots, v_k$ with associated subsets $S_1, \ldots, S_k$ of $\mathcal{S}$) or slack with probabilities:

$$p(v_l) = \frac{Z_w^{\text{UB}}(S_l)}{Z_w^{\text{UB}}(S)} \qquad p(\text{slack}) = 1 - \frac{\sum_{l=1}^k Z_w^{\text{UB}}(S_l)}{Z_w^{\text{UB}}(S)}$$

3. If a child was sampled with an associated subset containing a single element then return this element.
4. If a child, $v_l$, was sampled with an associated subset containing more than one element then set $v = v_l$, $S = S_l$, and go to step 2.
5. If the slack element was sampled then go to step 1.

---

## A.2 Adaptive Rejection Sampling

We can improve the efficiency of ADAPART by tightening the upper bounds $Z_w^{\text{UB}}$ whenever we encounter slack. This is done by subtracting the computed slack from the associated upper bounds, which still preserves nesting properties. The resulting algorithm is an *adaptive rejection sampler* [19], where the "envelope" proposal is tightened every time a point is rejected.[4]

428   Formally, for any partition $P$ of $S$, we define a new, tighter upper bound as follows:

$$\underline{Z}_w^{\text{UB}}(S) = \min\left\{\sum_{S_i \in P} Z_w^{\text{UB}}(S_i),\ Z_w^{\text{UB}}(S)\right\}. \tag{4}$$

429 This is still a valid upper bound on $Z_w^{\text{UB}}(S)$ because of the additive nature of $Z_w$, and is, by definition,
430 also nesting w.r.t. the partition $P$. If we encounter any slack, there must exists some $S$ for which
431 $\underline{Z}_w^{\text{UB}}(S) < Z_w^{\text{UB}}(S)$, hence we can *strictly* tighten our bound for subsequent steps of the algorithm
432 (thereby making ADAPART more efficient) by using $\underline{Z}_w^{\text{UB}}(S)$ instead of $Z_w^{\text{UB}}(S)$. Empirically we
433 find that bound tightening is most effective for small matrices. Sampling matrices uniformly from
434 $[0, 1)$, we find that after 1000 samples we improve our bound on the permanent to roughly 64%, 77%,
435 and 89% of the original bound for matrices of size 10, 15, 25 respectively. Bound tightening may be
436 more effective for other types of large matrices.

### A.3   Estimating the Partition Function with Adaptive Rejection Sampling

438 The number of accepted samples, $a$, is a random variable with expectation $E[a] = \sum_{i=1}^T \frac{Z}{Z_i^{UB}}$, where
439 $Z_i^{UB}$ is the upper bound on the entire state space $\mathcal{S}$ when the $i$-th sample is drawn. This gives the
440 unbiased estimator $\hat{Z} = a / \left(\sum_{i=1}^T \frac{1}{Z_i^{UB}}\right)$ for the partition function. We use bootstrap techniques
441 [12] to perform Monte Carlo simulations that yield high probability bounds on the partition function.

### A.4   Runtime Guarantee of ADAPART

443 Law [32] prove that the runtime of Algorithm A.1 is $O(n^{1.5 + .5/(2\gamma - 1)})$ per sample when using
444 their upper bound on the permanent [32, p. 33], where $\gamma$ controls density. ADAPART has the
445 same guarantee with a minor modification to the presentation in Algorithm 1. The repeat looped
446 is removed and if the terminating condition $ub \leq Z_w^{\text{UB}}(\mathcal{S})$ is not met after a single call to *Refine*,
447 Algorithm A.1 is called with the upper bound from and fixed partitioning strategy from [32] as shown
448 in Algorithm A.2.

### A.5   Additional Experiments

Figure 5: Accuracy results on randomly sampled $n$x$n$ block diagonal matrices constructed as described earlier, with blocks of size $k = 10$. We plot the exact permanent, our estimate, and our high probability bounds calculated from 10 samples for each matrix.

450 While calculating the permanent of a large matrix is generally intractable, it can be done efficiently
451 for certain special types of matrices. One example is block diagonal matrices, where an $n$x$n$ matrix is

**Algorithm A.2** ADAPART: Sample from a Normalized Weight Function using Adaptive Partitioning with Polynomial Runtime Guarantee for Dense Matrices

**Inputs:**

1. Non-empty state space $\mathcal{S}$
2. Unnormalized weight function $w : \mathcal{S} \to \mathbb{R}_{\geq 0}$
3. Family of upper bounds $Z_w^{\mathrm{UB}}(S) : \mathcal{D} \subseteq 2^{\mathcal{S}} \to \mathbb{R}_{\geq 0}$ for $w$ that are tight on single element subsets
4. Refinement function $Refine : \mathcal{P} \to 2^{\mathcal{P}}$ where $\mathcal{P}$ is the set of all partitions of $\mathcal{S}$

**Output:** A sample $i \in \mathcal{S}$ distributed as $i \sim \frac{w(i)}{\sum_{i \in \mathcal{S}} w(i)}$.

**if** $\mathcal{S} = \{a\}$ **then** Return $a$
$ub \leftarrow Z_w^{\mathrm{UB}}(\mathcal{S})$
$\{\{S_1^i, \cdots, S_{\ell_i}^i\}\}_{i=1}^K \leftarrow Refine(\mathcal{S})$
**for all** $i \in \{1, \cdots, K\}$ **do**
   $ub_i \leftarrow \sum_{j=1}^{\ell_i} Z_w^{\mathrm{UB}}(S_j^i)$
$j \leftarrow \arg\min_i ub_i$
$P \leftarrow \{S_1^j, \cdots, S_{\ell_j}^j\}$
$ub \leftarrow ub - Z_w^{\mathrm{UB}}(\mathcal{S}) + ub_j$
**if** $ub > Z_w^{\mathrm{UB}}(\mathcal{S})$ **then**
   Return the output of Algorithm A.1 called on $\mathcal{S}$ and $w$ with the bound and fixed partition of [32]
**else**
   Sample a subset $S_i \in P$ with prob. $\frac{Z_w^{\mathrm{UB}}(S_i)}{Z_w^{\mathrm{UB}}(\mathcal{S})}$, or sample $slack$ with prob. $1 - \frac{ub}{Z_w^{\mathrm{UB}}(\mathcal{S})}$
**if** $S_m \in P$ is sampled **then**
   Recursively call ADAPART $(S_m, w, Z_w^{\mathrm{UB}}, Refine)$
**else**
   Restart, i.e., call ADAPART $(\mathcal{S}, w, Z_w^{\mathrm{UB}}, Refine)$

---

452 composed of $\lfloor \frac{n}{k} \rfloor$ blocks of size $k$x$k$ and a single $n \bmod k$ block along the diagonal. Only elements
453 within these blocks on the diagonal may be non-zero. The permanent of a block diagonal matrix is
454 simply the product of the permanents of each matrix along the diagonal, which can be calculated
455 efficiently whenever the block size is sufficiently small. We plot the exact permanent, our estimate,
456 and our high probability bounds for randomly sampled block diagonal matrices of various sizes in
457 Figure 5.

### A.6 Multi-Target Tracking Overview

459 The multi-target tracking problem is very similar to classical inference problems in hidden Markov
460 models, requiring the estimation of an unobserved state given a time series of noisy measurements.
461 The non-standard catch is that at each time step the observer is given one noisy measurement per
462 target, but is not told which target produced which measurement. Brute forcing the problem is
463 intractable because there are $K!$ potential associations when tracking $K$ targets. The connection
464 between measurement association and the matrix permanent arises frequently in tracking literature
465 [45, 35, 36, 38], and its computational complexity is cited when discussing the difficulty of multi-
466 target tracking.

467 As brief background, the computational complexity of multi-target tracking has led to many heuristic
468 approximations, notably including multiple hypothesis tracking (MHT) [37, 15, 31] and joint proba-
469 bilistic data association (JPDA) [18, 38]. As heuristics, they can succumb to failure modes. JPDA is
470 known to suffer from target coalescence where neighboring tracks merge [9].

471 Alternatively, sequential Monte Carlo methods (SMC or particle filters) provide an asymptotically
472 unbiased method for sequentially sampling from arbitrarily complex distributions. When targets
473 follow linear Gaussian dynamics, a Rao-Blackwellized particle filter may be used to sample the
474 measurement associations allowing sufficient statistics for distributions over individual target states
475 to be computed in closed form (by Kalman filtering, see Algorithm A.3 in the Appendix for further
476 details) [41]. The proposal distribution is a primary limitation when using Monte Carlo methods.

<sup>477</sup> Ideally it should match the target distribution as closely as possible, but this generally makes it
<sup>478</sup> computationally unwieldy.

<sup>479</sup> In the case of a Rao-Blackwellized particle filter for multi-target tracking, the optimal proposal
<sup>480</sup> distribution [17, p. 199] that minimizes the variance of each importance weight is a distribution
<sup>481</sup> over permutations defined by a matrix permanent (please see Section A.10 in the Appendix for
<sup>482</sup> further details). We implemented a Rao-Blackwellized particle filter that uses the optimal proposal
<sup>483</sup> distribution. We evaluated it's effectiveness against a Rao-Blackwellized particle filter using a
<sup>484</sup> sequential proposal distribution [41] and against the standard multiple hypothesis tracking framework
<sup>485</sup> (MHT) [37, 15, 31].

<sup>486</sup> Our work can be extended to deal with a variable number of targets and clutter measurements using a
<sup>487</sup> matrix formulation similar to that in [3].

## A.7 Optimal Single-Target Bayesian Filtering

<sup>489</sup> In this section we give a brief review of the optimal Bayesian filter for single-target tracking. Consider
<sup>490</sup> a hidden Markov model with unobserved state $\mathbf{x}_t$ and measurement $\mathbf{y}_t$ at time $t$. The joint distribution
<sup>491</sup> over states and measurements factors as

$$\Pr(\mathbf{x}_{1:T}, \mathbf{y}_{1:T}) = \Pr(\mathbf{x}_1)\Pr(\mathbf{y}_1|\mathbf{x}_1)\prod_{t=2}^{T}\Pr(\mathbf{x}_t|\mathbf{x}_{t-1})\Pr(\mathbf{y}_t|\mathbf{x}_t)$$

<sup>492</sup> by the Markov property. This factorization of the joint distribution facilitates Bayesian filtering,
<sup>493</sup> a recursive algorithm that maintains a fully Bayesian distribution over the hidden state $\mathbf{x}_t$ as each
<sup>494</sup> measurement $\mathbf{y}_t$ is sequentially observed. Given the prior distribution $p(\mathbf{x}_1)$ over the initial state, the
<sup>495</sup> Bayesian filter consists of the update step[5]

$$\Pr(\mathbf{x}_t|\mathbf{y}_{1:t}) = \frac{\Pr(\mathbf{y}_t|\mathbf{x}_t)\Pr(\mathbf{x}_t|\mathbf{y}_{1:t-1})}{\int \Pr(\mathbf{y}_t|\mathbf{x}_t)\Pr(\mathbf{x}_t|\mathbf{y}_{1:t-1})d\mathbf{x}_t}$$

<sup>496</sup> and the prediction step

$$\Pr(\mathbf{x}_t|\mathbf{y}_{1:t-1}) = \int \Pr(\mathbf{x}_t|\mathbf{x}_{t-1})\Pr(\mathbf{x}_{t-1}|\mathbf{y}_{1:t-1})d\mathbf{x}_{t-1}.$$

<sup>497</sup> In the special case of linear Gaussian models where the state transition and measurement processes
<sup>498</sup> are linear but corrupted with Gaussian noise, the above integrals can be computed analytically giving
<sup>499</sup> closed form update and predict steps. The distribution over the hidden states remains Gaussian and is
<sup>500</sup> given by the Kalman filter with update step

$$\Pr(\mathbf{x}_t|\mathbf{y}_{1:t}) = \mathcal{N}(\hat{\mathbf{x}}_{t|t}, \mathbf{P}_{t|t}) \tag{5}$$

<sup>501</sup> and prediction step

$$\Pr(\mathbf{x}_t|\mathbf{y}_{1:t-1}) = \mathcal{N}(\hat{\mathbf{x}}_{t|t-1}, \mathbf{P}_{t|t-1}). \tag{6}$$

## A.8 Optimal Multi-Target Bayesian Filtering

<sup>503</sup> In this section we give a brief review of the optimal Bayesian filter for multi-target tracking problem
<sup>504</sup> with a fixed cardinality (fixed number of targets and measurements over time) [36, pp. 485-486] and
<sup>505</sup> its computational intractability.

<sup>506</sup> Given standard multi-target tracking assumptions , the joint distribution over all target states $X$,
<sup>507</sup> measurements $Y$, and measurement-target associations $\pi$ can be factored as[6]

$$\Pr(X, Y, \pi) = \Pr(X_1)\Pr(\pi_1)\Pr(Y_1|X_1, \pi_1)$$
$$\times \prod_{t=2}^{T}\Pr(X_t|X_{t-1})\Pr(\pi_t)\Pr(Y_t|X_t, \pi_t). \tag{7}$$

The optimal Bayesian filter for multi-target tracking is a recursive algorithm, similar to the standard
Bayesian filter in the single target tracking setting, that maintains a distribution over the joint state
of all targets by incorporating new measurement information as it is obtained. It is more complex
than the single target Bayesian filter because it must deal with uncertainty in measurement-target
association. As in the single target tracking setting the filter is composed of prediction and update
steps. The prediction step is

$$
\begin{aligned}
&\Pr(X_t|Y_{1:t-1}) \\
&= \sum_{\pi_{1:t-1}} \Pr(X_t|Y_{1:t-1}, \pi_{1:t-1}) \Pr(\pi_{1:t-1}|Y_{1:t-1}) \\
&= \frac{1}{k!^{t-1}} \sum_{\pi_{1:t-1}} \Pr(X_t|Y_{1:t-1}, \pi_{1:t-1}) \\
&= \frac{1}{k!^{t-1}} \sum_{\pi_{1:t-1}} \Pr((X_t^1, \ldots, X_t^K)|Y_{1:t-1}, \pi_{1:t-1}) \\
&= \frac{1}{k!^{t-1}} \sum_{\pi_{1:t-1}} \int \cdots \int \Pr(X_t^1|X_{t-1}^1) \Pr(X_{t-1}^1|Y_{1:t-1}, \pi_{1:t-1}) \\
&\qquad \times \Pr(X_t^K|X_{t-1}^K) \Pr(X_{t-1}^K|Y_{1:t-1}, \pi_{1:t-1}) dX_{t-1}^1 \ldots dX_{t-1}^K \\
&= \frac{1}{k!^{t-1}} \sum_{\pi_{1:t-1}} \int \Pr(X_t^1|X_{t-1}^1) \Pr(X_{t-1}^1|Y_{1:t-1}, \pi_{1:t-1}) dX_{t-1}^1 \\
&\qquad \times \cdots \times \\
&\qquad \int \Pr(X_t^K|X_{t-1}^K) \Pr(X_{t-1}^K|Y_{1:t-1}, \pi_{1:t-1}) dX_{t-1}^K.
\end{aligned}
\tag{8}
$$

The update step is

$$
\begin{aligned}
&\Pr(X_t|Y_{1:t}) \\
&= \sum_{\pi_{1:t}} \Pr(X_t|Y_{1:t}, \pi_{1:t}) \Pr(\pi_{1:t}|Y_{1:t}) \\
&= \frac{1}{k!^t} \sum_{\pi_{1:t}} \Pr(X_t|Y_{1:t}, \pi_{1:t}) \\
&= \frac{1}{k!^t} \sum_{\pi_{1:t}} \frac{\Pr(Y_t|X_t, \pi_t) \Pr(X_t|Y_{1:t-1}, \pi_{1:t-1})}{\int \Pr(Y_t|X_t, \pi_t) \Pr(X_t|Y_{1:t-1}, \pi_{1:t-1}) dX_t}
\end{aligned}
\tag{9}
$$

Unfortunately the multi-target optimal Bayesian filtering steps outlined above are computationally
intractable to compute. Even in special cases where the integrals are tractable, such as for linear
Gaussian models, summation over $k!^t$ states is required.

### A.9 Sequential Monte Carlo

Sequential Monte Carlo (SMC) or particle filtering methods can be used to sample from sequential
models . These methods can be used to sample from the distribution defined by the optimal Bayesian
multi-target filter. When target dynamics are linear Gaussian a Rao-Blackwellized particle filter can
be used to sample measurement-target associations and compute sufficient statistics for individual
target distributions in closed form [41].

Pseudo-code for Rao-Blackwellized sequential importance sampling is given in algorithm A.3. We
use $KF_u(\cdot)$ and $KF_p(\cdot)$ to denote calculation of the closed form Kalman filter update and prediction
steps given in equations 5 and 6 respectively.

### A.10 Optimal Proposal Distribution

While SMC methods are asymptotically unbiased, their performance depends on the quality of the pro-
posal distribution. The optimal proposal distribution that minimizes the variance of importance weight

Rao-Blackwellized Sequential Importance Sampling

**Outputs:** $N$ importance samples $\pi_{1:T}^{(i)} \sim \Pr(\pi_{1:T}|Y_{1:T})$ and weights $w_T^{(i)}$ ($i \in 1, 2, \ldots, n$) with corresponding state estimates $\hat{X}_{1:T}^{(i)}$ and covariance matrices $P_{1:T}^{(i)}$. Note $\hat{X}_{1:T}^{(i)}$ and $P_{1:T}^{(i)}$ are both arrays; $\hat{X}_t^{k(i)}$ is the $k^{\text{th}}$ target's estimated state vector at time $t$ for sample $i$.

1:    **for** t = 1, ..., T **do**   // Update particle at time $t$
2:      **for** i = 1, ..., N **do**// Sample particle i
3:         $\pi_t^{(i)} \sim q(\pi_t|\pi_{1:t-1}^{(i)}, Y_{1:t})$
4:         $\pi_{1:t}^{(i)} \leftarrow \left(\pi_{1:t-1}^{(i)}, \pi_t^{(i)}\right)$
5:         **for** k = 1, ..., K **do**// Iterate over targets
6:            $\hat{X}_{t|t}^{k(i)}, P_{t|t}^{k(i)} \leftarrow KF_u\left(\hat{X}_{t|t-1}^{k(i)}, P_{t|t-1}^{k(i)}, Y_t^{\pi^{(k)}}\right)$
7:            $\hat{X}_{t+1|t}^{k(i)}, P_{t+1|t}^{k(i)} \leftarrow KF_p\left(\hat{X}_{t|t}^{k(i)}, P_{t|t}^{k(i)}\right)$
8:            $\hat{X}_{1:t}^{(i)} \leftarrow \left(\hat{X}_{1:t-1}^{(i)}, \hat{X}_t^{(i)}\right)$
9:            $P_{1:t}^{(i)} \leftarrow \left(P_{1:t-1}^{(i)}, P_t^{(i)}\right)$
10:        $w_t^{*(i)} \leftarrow w_{t-1}^{*(i)} \frac{\prod_{k=1}^{K} P\left(Y_{1:t}^{\pi_t(k)}|\hat{X}_{t|t-1}^{k(i)}, P_{t|t-1}^{k(i)}\right)}{q(\pi_t|\pi_{1:t-1}^{(i)}, Y_{1:t})}$
11:     **for** i = 1, ..., N **do**// Normalize importance weights
12:        $\tilde{w}_t^{(i)} \leftarrow \frac{w_t^{*(i)}}{\sum_{j=1}^{N} w_t^{*(j)}}$
13:

530   $w_t^{*(i)}$ [17, p. 199] is $q(x_t|x_{1:t-1}^{(i)}, Y_{1:t}) = \Pr(x_t|x_{t-1}^{(i)}, Y_t)$. In our setting we have hidden variables $X$
531   and $\pi$, so we need to rewrite this as $q(X_t, \pi_t|X_{1:t-1}^{(i)}, \pi_{1:t-1}^{(i)}, Y_{1:t}) = \Pr(X_t, \pi_t|X_{t-1}^{(i)}, \pi_{t-1}^{(i)}, Y_t) =$
532   $\Pr(X_t, \pi_t|X_{t-1}^{(i)}, Y_t)$ (note that $X_t$ and $\pi_t$ are conditionally independent from $\pi_{t-1}^{(i)}$ given $X_{t-1}^{(i)}$).
533   Using Rao-Blackwellization we avoid sampling $X_t$ but instead compute sufficient statistics (mean
534   and covariance) in closed form, so we have that the optimal proposal distribution is

$$
\begin{aligned}
& q(\pi_t|X_{1:t-1}^{(i)}, \pi_{1:t-1}^{(i)}, Y_{1:t}) \\
&= \Pr(\pi_t|\hat{X}_{t|t-1}^{(i)}, P_{t|t-1}^{(i)}, \pi_{1:t-1}^{(i)}, Y_{1:t}) \\
&= \Pr(\pi_t|\hat{X}_{t|t-1}^{(i)}, P_{t|t-1}^{(i)}, Y_{1:t}) \\
&= \frac{\Pr(\pi_t, \hat{X}_{t|t-1}^{(i)}, P_{t|t-1}^{(i)}, Y_{1:t})}{\sum_{\pi_t} \Pr(\pi_t, \hat{X}_{t|t-1}^{(i)}, P_{t|t-1}^{(i)}, Y_{1:t})} \\
&= \frac{\Pr(Y_{1:t}|\pi_t, \hat{X}_{t|t-1}^{(i)}, P_{t|t-1}^{(i)}) \Pr(\hat{X}_{t|t-1}^{(i)}, P_{t|t-1}^{(i)}|\pi_t) \Pr(\pi_t)}{\sum_{\pi_t} \Pr(Y_{1:t}|\pi_t, \hat{X}_{t|t-1}^{(i)}, P_{t|t-1}^{(i)}) \Pr(\hat{X}_{t|t-1}^{(i)}, P_{t|t-1}^{(i)}|\pi_t) \Pr(\pi_t)} \\
&= \frac{\Pr(Y_{1:t}|\pi_t, \hat{X}_{t|t-1}^{(i)}, P_{t|t-1}^{(i)}) \Pr(\hat{X}_{t|t-1}^{(i)}, P_{t|t-1}^{(i)}) \Pr(\pi_t)}{\sum_{\pi_t} \Pr(Y_{1:t}|\pi_t, \hat{X}_{t|t-1}^{(i)}, P_{t|t-1}^{(i)}) \Pr(\hat{X}_{t|t-1}^{(i)}, P_{t|t-1}^{(i)}) \Pr(\pi_t)} \\
&= \frac{\Pr(Y_{1:t}|\pi_t, \hat{X}_{t|t-1}^{(i)}, P_{t|t-1}^{(i)})/k!}{\sum_{\pi_t} \Pr(Y_{1:t}|\pi_t, \hat{X}_{t|t-1}^{(i)}, P_{t|t-1}^{(i)})/k!} \\
&= \frac{\prod_{k=1}^{K} \Pr(Y_{1:t}^{\pi_t(k)}|\hat{X}_{t|t-1}^{k(i)}, P_{t|t-1}^{k(i)})}{\sum_{\pi_t} \prod_{k=1}^{K} \Pr(Y_{1:t}^{\pi_t(k)}|\hat{X}_{t|t-1}^{k(i)}, P_{t|t-1}^{k(i)})}.
\end{aligned}
\tag{10}
$$

Note that the denominator of the final line in equations 10 is the permanent of matrix $A$, where $(a_{jk}) = \Pr(Y_{1:t}^{j} | \hat{X}_{t|t-1}^{k(i)}, P_{t|t-1}^{k(i)})$. Using the machinery developed throughout this paper we can sample from the optimal proposal distribution and compute approximate importance weights .

## Footnotes

[4]The use of 'adaptive' here is to connect this section with the rejection sampling literature, and is unrelated to 'adaptive' partitioning discussed earlier.

[5]Where we have abused notation and the initial distribution is $\Pr(\mathbf{x}_1|\mathbf{y}_{1:0}) = \Pr(\mathbf{x}_1)$.

[6]For a tracking sequence of $K$ targets over $T$ time steps, $X$ is an array where row $X_t = (X_t^1, \ldots, X_t^K)$ represents the state of all targets at time $t$ and element $X_t^k$ is a vector representing the state of the $k^{\text{th}}$ target at time $t$. Likewise $Y$ is an array where row $Y_t = (Y_t^1, \ldots, Y_t^K)$ represents all measurements at time $t$ and element $Y_t^k$ is a vector representing the $k^{\text{th}}$ measurement at time $t$. Measurement-target associations are represented by the array $\pi$ where the element $\pi_t \in S_k$ is a permutations of $\{1, 2, \ldots, k\}$ ($S_k$ denotes the symmetric group).