[Reviews · NeurIPS 2019]

Reviewer 1



The problem of computing the permanent of a non-negative matrix is probably the best known example of a #P complete problem, in addition to being a useful tool in graph theory, statistics, and increasingly in quantum computation with respect to boson sampling. The main theoretical contribution of this paper lies in replacing the original workflow of Huber and Law for exact partition sampling - which required an proof in advance that a particular upper bound satisfied the nesting property - with an adaptive method that works for any upper bound. This is a significant result given the difficulty of applying the original work and the accompanying implementation is likely to become widely used. The paper is well written and the explanation of the problem setting (existing FPRAS method that no one actually implements) reflects the current state of the field. The new ideas presented in Section 3 are intuitive and well motivated. The derivation choices for the specific case of the permanent seem natural, although a comparison of the results using different bounds would be interesting, since the dependence of the run time on choice of bounds is not explored. The experiments are reasonably comprehensive - larger matrices without the block diagonal structure could have been constructed by taking sequences of combinatorial graphs for which the number of cycle covers is known. While the block diagonal matrices are sparse in the formal sense, it isn't clear that they are representative of non-dense matrices more generally. Overall, this paper presents an important new idea and provides more than sufficient validation of its practical implications. *Smaller Notes In the introduction, it is the bi-adjacency matrix (where the rows and columns are separately indexed by the partite sets) whose permanent counts perfect matchings. I found the informal description of the nesting property in the contributions section a little confusing although the formal definitions are clear. It would be nice to see Figure 5 from the supplement in the main text but page restrictions may make that difficult. (very) small note, but many places with multiple references they are out of (numerical) order. I appreciate the authors responses and understand the computational difficulties involved in actually applying the other types of bounds. In my mind this does not detract from the strength of the paper.

Reviewer 2



The authors propose AdaPart, an adaptive partitioning scheme for a rejection sampling strategy for estimating the permanent. The novelty over existing work is the introduction of a greedy method to build the partition trees on the fly (empirical verification shows that this approach is quite effective in practice) along with an application of a tighter upper bound (which reduces both the practical and theoretical cost of this type of sampling scheme). The paper is well-written and careful to place its contributions with respect to existing work on permanent approximation. My only criticism, really, is that the approach seems to be much more applicable than just a strategy for computing the permanent and it would have been nice to see some experiments on different types of partition function estimation tasks.

Reviewer 3



In overall, I think this paper is both significant and practically useful for estimating the partition function. I enjoyed reading the paper and think it is novel enough to be accepted. Experiments are convincing and well designed. However, I am slightly concerned about the generality of the algorithm. Although the algorithm was written in a general way, i.e., not tied to the permanent problem and particular choice of bound, only a single choice of upper bound from Soules is considered for the permanent problem. This is potentially problematic since the algorithm might result in too many refinements and inapplicable for other upper bounds, unlike the claims that the authors made. I also think the writing can be improved. The term "upper bound" and "nested upper bound" was confusing when I was first reading the paper. Specifically, I would like to suggest clarifying that "it is provided that Z_{w}^{UB} is always an upper bound for Z_{w}", while "it is non-trivial to see that Z_{w}^{UB} is nesting upper bound". Also, I think Figure 1 is slightly confusing since we are discarding the unused partitions as we go down the node hierarchy, but the figure seems to require 2^{n} choice of partitions. minor: - I think it would be empirically interesting to see how much the upper bound is likely to fail in nesting (for a single partition). I suspect there might be extreme cases where nesting the upper bound is quite hard than others.

[Author Response · NeurIPS 2019]

We thank all reviewers for their time and thoughtful comments.

# 1 Response to Reviewer 1

*It would also be nice to see some discussion of how the given procedure would have to be modified (or how difficult such a modification should be expected to be) to incorporate some of the other bounds discussed in the paper:* Writing the code to incorporate additional bounds is actually very easy. For instance, one can define a new upper bounding function that returns the minimum of two other upper bounding functions. The difficulty is in finding other upper bounding functions that are (1) tighter than the bound we use, (2) efficient to compute, and (3) empirically require few calls to *Refine*. We did not find other upper bounds with implementations that satisfied conditions (1) and (2). We would be happy to include a discussion of bounds that we believe are promising to explore. Particularly, we think that the Bethe upper bound could yield improvements on sparse matrices (as in Fig. 3), which could be useful for multi-target tracking where the matrix of interest is frequently sparse. Unfortunately, the fast c implementation of the bound provided by [1] is numerically unstable for sparse matrices with 0 entries and the numerically stable matlab implementation is prohibitively slow. This difficulty could be overcome by rewriting an efficient implementation. Another potentially interesting bound to explore is the "sharpened" version of the bound we use, described in [4]. This bound is computed by solving an optimization problem, but unfortunately we do not know of an efficient solution. Using gradient descent, we found that this sharpened bound can be significantly tighter than the one we use, but this approach is too computationally expensive. We believe an efficient solution may exist, but have not found it.

*It would be nice to see some comparison of the estimates of the values of the previous Law sampling method in addition to the runtime comparison:* Both our method and Law's method provide exact samples, so we would obtain the same estimates up to random effects.

We appreciate the additional suggestions to improve our paper.

# 2 Response to Reviewer 2

*My only criticism, really, is that the approach seems to be much more applicable than just a strategy for computing the permanent and it would have been nice to see some experiments on different types of partition function estimation tasks:* We believe that implementing ADAPART to work for general graphical models is indeed an interesting direction for future work. We would like to explore implementation using a variety of general upper bounds [5, 2, 3].

# 3 Response to Reviewer 3

*Although the algorithm was written in a general way, i.e., not tied to the permanent problem and particular choice of bound, only a single choice of upper bound from Soules is considered for the permanent problem:* Please refer to our discussion in response to reviewer 1.

Thank you for your suggestions, we will clarify Figure 1 and our use of the term "nested upper bound."

# References

[1] B. Huang and T. Jebara. Approximating the permanent with belief propagation. *arXiv preprint arXiv:0908.1769*, 2009.

[2] Q. Liu and A. T. Ihler. Bounding the partition function using holder's inequality. In *ICML*, 2011.

[3] Q. Lou, R. Dechter, and A. Ihler. Anytime anyspace and/or search for bounding the partition function. In *AAAI*, 2017.

[4] G. W. Soules. Permanental bounds for nonnegative matrices via decomposition. *Linear algebra and its applications*, 394:73–89, 2005.

[5] M. J. Wainwright, T. S. Jaakkola, and A. S. Willsky. Tree-reweighted belief propagation algorithms and approximate ml estimation by pseudo-moment matching. In *AISTATS*, 2003.


[Meta-Review · NeurIPS 2019]

Reviewers quite liked this contribution, rating it significant for both practical and theoretical contributions, with few complaints except a desire to see the method made even more general. Hopefully the reviewer suggestions will be helpful to the authors. We look forward to seeing the final version.